# Benefits of Cultural Activities on People with Cognitive Impairment: A Meta-Analysis

**DOI:** 10.3390/healthcare11131854

**Published:** 2023-06-26

**Authors:** Laia Delfa-Lobato, Maria Feliu-Torruella, Cristina Cañete-Massé, Silvia Ruiz-Torras, Joan Guàrdia-Olmos

**Affiliations:** 1Department of Applied Didactics, Faculty of Education, University of Barcelona, 08035 Barcelona, Spain; mfeliu@ub.edu; 2Institute of Research in Education (IRE), University of Barcelona, 08035 Barcelona, Spain; 3Department of Social Psychology and Quantitative Psychology, Faculty of Psychology, University of Barcelona, 08035 Barcelona, Spain; cristinacanete@ub.edu (C.C.-M.); jguardia@ub.edu (J.G.-O.); 4UB Institute of Complex Systems, University of Barcelona, 08035 Barcelona, Spain; 5Psychological Clinic, Josep Finestres Foundation, Faculty of Psychology, University of Barcelona, 08035 Barcelona, Spain; silvia.ruiz@ub.edu; 6Institute of Neuroscience, University of Barcelona, 08035 Barcelona, Spain

**Keywords:** meta-analysis, cognitive impairment, cultural activities, Alzheimer’s disease, dementia, arts and health

## Abstract

Background: Museums and cultural institutions are increasingly aware of both the interests and needs of society. Accordingly, these institutions are becoming allies in terms of health and well-being due to the importance of their social functions. Presently, many institutions create cultural activities aimed at cognitively impaired people, a group on the rise owing to the prevalence of dementia and the aging of society. Nevertheless, scientific evidence in this field remains scarce. As a result, the main objective of this research was to empirically evaluate and identify the benefits that cultural interventions can bring to cognitively impaired participants. Method: A meta-analysis (MA) was performed following PRISMA guidelines. When inclusion and eligibility criteria had been established, articles were subsequently selected through a strategic search of Web of Science, SCOPUS, PubMed, and Medline. Results: Twenty-six studies met the eligibility criteria, involving a total of 1201 participants with cognitive impairment. The results showed a statistically non-significant effect size when analyzing these cultural interventions for cognitively impaired people overall. However, when conducting partial meta-analyses (MA’), focusing on studies related to a specific disease, a particular type of treatment, or a specific type of evaluation, the results concurred with the conclusion of the previous systematic review (SR). Conclusion: Despite the high heterogeneity of the studies, benefits were identified in emotional well-being and social aspects but not in clinical ones such as the deterioration of cognitive or motor function, among others.

## 1. Introduction

Cognitive impairment (CI) is a condition that occurs concomitantly with various diseases [1]. It is the most common cause of CI Dementia of the Alzheimer’s Type (DAT) and is a syndrome characterized by impairing cognitive function [2], which affects more than half of the people living with CI [3,4]. It is also important to highlight that so-called vascular dementia is the second cause of CI, responsible for between 25% and 30% of cases [3]. CI is considered one of the most significant conditions affecting the older population, as it is associated with memory loss, difficulties remembering, concentration, or learning new things, as well as provoking feelings of confusion [5].

Dementias are currently affecting 50 million people [2], and the World Health Organization (WHO) forecasts that by 2030, the number of people living with dementia will rise to almost 152 million.

It is important to mention that dementia does not only affect those suffering directly from it; there is also a huge socioeconomic impact, particularly on counties’ healthcare systems as well as on those families caring for someone with that syndrome [6].

Cultural activities have proven to be beneficial for people living with Parkinson’s Disease (PD) [7,8] and other CI causes [9,10]. The benefits of arts and cultural activities can include increasing their well-being, helping with their social interaction and communication, and improving their QoL [11,12].

To delimit cultural activities, we aim to define them as various forms of human expression, engagement, and interaction that originate from a particular culture, encompassing its traditions, beliefs, values, customs, arts, and intellectual achievements [13]. In the context of the Throsby model, these activities are connected to core artistic expressions and publicly funded cultural institutions [14].

In recent decades, the cultural and heritage sectors have been increasingly providing spaces where people can go to find well-being and Quality of Life (QoL) [15]. Thanks to changing conceptions, the community has become progressively aware of what these institutions can provide for them in terms of meeting their interests and, above all, their needs [16,17]. In this way, museums and cultural institutions are proving to be competent and powerful allies for public health and wellness programs [15,18]. 

The interest in the culture and heritage sector and its implications in terms of public health and well-being has increased during the last few years, making researchers from other areas, institutions, and governments take this sector into account [19,20]. Nevertheless, there is still a lack of solid scientific evidence to prove the effectiveness of cultural practices in terms of health and well-being in general and specifically with this type of practice aimed at people living with a CI condition [21]. However, since the main cause of CI is still incurable, anything and everything that can be beneficial for people living with it must be taken into consideration.

This investigation, therefore, sets out to prove the effectiveness of cultural practices on health and well-being. It is a follow-on to a previous SR [22] centered on cultural practice directed at people living with CI, and it hopes to counterbalance the current lack of meta-analysis (MA) focused on this topic. The main aim of this research was to locate and analyze studies based on cultural activities to evaluate, identify, and describe their benefits for cognitively impaired people who participated independently of the cause of their condition. To conduct the MA properly, PRISMA guidelines [23,24] were followed. 

## 2. Materials and Methods

### 2.1. Study Eligibility

The inclusion criteria for this MA included (1) published empirical studies about arts and culture as health and well-being activities for people diagnosed with initial- or middle-stage CI; (2) studies in English, French, Spanish, or Catalan; (3) papers published between 2010 and 2021 in order to analyze the newest resources, discarding the previous studies as outdated; (4) studies centered on interventions adjusted for groups; (5) with active or passive engagement of the participants in art and culture activities (i.e., the creation of something artistic or hearing, seeing, or touching artistic/cultural elements); (6) cultural activities taking place in a museum or in a cultural environment, or not; (7) interventions oriented to cognitively impaired diagnosed people, cognitively impaired diagnosed people and their caregivers, and/or cognitively impaired diagnosed people and their family members; (8) quantitative or mixed study designs; (9) studies including pre-test and post-test measures, reporting standard deviation (SD) and mean or data that allow to calculate it; and (10) reports must include a control or comparison group, providing the total number of groups as well as the number of participants in each one.

Population: people diagnosed with CI.

Intervention: cultural and art-based interventions corresponding to UNESCO’s culture definition and framed within Throsby’s cultural industries model [14,25].

Comparators: only studies with an appropriate control or comparison group were included. Various treatment designs were included, all involving cultural treatment. Cultural treatment encompasses activities such as visiting museums, engaging in visual arts, listening to or performing music, dancing or watching dance performances, participating in performing arts, creating ceramics, attending readings or writing, or a combination of these activities. The treatment designs considered were cultural treatment compared to control, cultural treatment compared to another treatment type, and comparisons within different cultural treatment groups. Included studies must include pre-test and post-test measures and reports and must specify mean and SD measures in the pre-test and post-testOutcome: quantitative measures of intervention results.

Study type: interventional clinical trials (ICT).

Reports were excluded if (1) they were defined as dissertations, books, book chapters, MA or SR, individual interventions, or non-empirical studies; (2) they were studies of cultural activities that did not correspond to those falling under UNESCO’s culture definition [25] and were also framed within the cultural industries model defined by Throsby [14].

### 2.2. Search Strategy 

To identify suitable papers for this review, a search was conducted independently by one of the authors (L.D.L.) and an independent reviewer. We obtained a 100% rate of agreement between the two investigators for the study search and selection. The examined databases were Web of Science (WOS), SCOPUS, PubMed, and Medline. The search was restricted to 2010–2021.

The search was conducted in June 2021. Key search terms used were (muse* OR art OR “heritage site” OR “cultural engagement”). Previous words were employed in combination with the terms referring to CI (Alzheimer OR “CI” OR dement* OR “cognitive disfunction” OR “cognitive decline” OR “mild cognitive impairment”), using proximity Boolean search operators such as NEAR or W/n (*n* = 100) instead of AND when the database permitted it. To refine the search, words concerning artistic and cultural activities were included, as well as those concepts related to arts and health or art therapy (“arts and well-being” OR “arts and humanities” OR “arts and health” OR “reminiscence therapy” OR “art therapy” OR “dance therapy” OR “music therapy” OR “singing” OR “performing art” OR “theater” OR “cinema” OR “life story” OR “life review” OR “storytelling” OR “visual art” OR “creative art” OR “paint” OR “painting” OR “drawing” OR “collage” OR “pottery” OR “sculpture” OR “contemporary art” OR “art gallery” OR “photography”). For WOS and SCOPUS, the search was applied to all databases. A search procedure was built within the Medline database using Medical Subject Headings (MeSH), following the PICO strategy [26] to optimize and adapt the search question and the search strategy used in the rest of the databases.

### 2.3. Data Extraction and Variable Coding

Initially, data were extracted and coded by one reviewer (L.D.-L.) using a data extraction form precisely designed and agreed upon by all the authors to reach these MA objectives. After that, an external reviewer checked the data extraction to ensure accuracy. Furthermore, the five authors (L.D.-L., M.F.-T., C.C.-M., S.R.-T., and J.G.-O.) discussed and settled any discrepancies. The data extraction form included the following information:

General: designed reference, including first author and publication year.

Participants: Type of cognitive impairment: Alzheimer’s, Parkinson’s, dementia, mild cognitive impairment, or aging (older adults over 60 [27,28]). The mean age of the participant groups was also included.

Treatment and study: number of groups participating, number of people in each group. Treatment type (treatment intervention type): music, dance, dance and music, reminiscence, visual arts, scenic arts. All the interventions were delivered in a group format. Treatment design: cultural vs. cultural vs. control, cultural vs. cultural, cultural vs. other treatment type, cultural vs. other treatment type vs. control, or cultural vs. control. Study design: quasi-randomized or randomized. Cultural treatment involves participating in various cultural experiences, such as visiting museums, engaging in visual arts (e.g., painting, drawing), enjoying music through listening or performing, experiencing dance performances, participating in performing arts (e.g., theater, opera), engaging in ceramics or pottery creation, attending readings or literary events, and engaging in writing activities.

Outcomes: indicator and its mean and SD pre-test and post-test for control and intervention groups, as well as its category and subcategory.

### 2.4. Effect Size Index/s and Statistical Analysis

In order to compare the results of the included interventions and to obtain general conclusions based on empirical data, the effect size was obtained using the correlation coefficient r and Cohen’s d, which are two of the most commonly used indexes to compare differences between means [29]. A Cohen’s d of 0.20 is classified as a small effect size, while a value of 0.5 is considered medium, and a value of 0.8 is considered large [30]. Effect size calculation aims to facilitate the combination and comparison of results from different interventions as well as indicate the intensity of the statistical effect observed [31].

In statistical analysis, the *p*-value and confidence interval are used to assess significance. When evaluating heterogeneity in meta-analysis, the Q statistic measures total variance [32], and the I^2^ index quantifies the proportion of variation due to true heterogeneity rather than chance [31,33].

### 2.5. Quality Appraisal

Randomized Controlled Trials and Quasi-randomized trials (non-randomized studies) were included in this MA.

As trials with two different types of sample selection were included, two different appraisal tools were used, choosing in each case the most appropriate from the ones provided by the JBI Sumari software (https://sumari.jbi.global/). For Randomized Controlled Trials, the JBI critical appraisal checklist for randomized controlled trials was used, while the JBI Critical Appraisal Checklist for Quasi-Experimental Studies was used for Quasi-Experimental trials (see Appendix A).

Randomized Controlled Trials have a weakness in terms of participants and treatment providers being aware of who receives the treatment. Assessors, however, have a higher level of ignorance. Group allocation concealment also has room for improvement.

In Quasi-Experimental trials, the main weakness is the lack of information about participants receiving similar treatment or care, with only a small number of studies addressing this. Other aspects of evaluation for Quasi-Experimental trials have higher scores.

## 3. Results

### 3.1. Search Outcome

A total of 1202 documents were identified from the search, of which 342 were duplicates. A total of 830 records were screened by title and abstract, and 675 were removed because they did not meet the inclusion criteria. A total of 154 articles were full-text screened, of which 26 met the inclusion criteria and were enrolled in the study (Figure 1). The included studies are marked with an * in the reference list.

### 3.2. Study Characteristics

The 26 included studies, published between 2011 and 2021, involved a total of 1201 participants. The studies were conducted in various countries, including Australia [34,35], Brazil [36], Canada [37,38], China [39,40,41,42], Denmark [43], France [44,45], Greece [46], Italy [7,47,48], Japan [49,50,51], the Republic of China [52], Spain [53], and the United States [10,54,55,56,57]. Most publications were in English, with one study in Spanish. The majority of the studies divided participants into two groups, some into three, and one into four. Different types of treatments based on cultural activities were compared for individuals with various types of cognitive impairment, with dance being the most common cultural intervention type. The included studies covered various types of cognitive impairment. The characteristics of the included studies are summarized in Table 1.

### 3.3. Benefits Analysis

#### 3.3.1. General Results

Due to the lack of empirical evidence related to the effect size behavior and whether it would behave the same in all the studies or not, it was initially decided to carry out the analysis for both the equal-effects model and the random-effects model.

For the equal-effects model, the obtained effect size was *ω* = −0.00, CI [−0.02, 0.02], and *p* = 0.82, showing a null effect. The results obtained for the *Q*-test and *I*^2^ demonstrate very high heterogeneity (*Q* = 2865.47, *p* < 0.01, *I*^2^ = 90.30%).

The obtained effect size for the random-effects model was *ω* = −0.02, CI [−0.13, 0.09], and *p* = 0.69, showing a null effect. Regarding the heterogeneity, the obtained results indicate that it was very high (*Q* = 2878.94, *p* < 0.01, *I*^2^ = 96.92%).

As the meta-analysis results obtained when analyzing the aggregated data from the 26 included studies were not statistically significant, a segmented analysis using partial meta-analyses (MA’) was carried out. The objective was to better understand how and when cultural interventions can be beneficial for people living with CI, as some benefits were detected in the previous systematic review.

Being conscious of the existing heterogeneity, in this case only the random-effect model was used to perform the partial MA’.

#### 3.3.2. Analysis per Treatment

Dance: The obtained effect size for dance as a treatment type was *ω* = 0.01, CI [0.08, 0.10], *p* = 0.79, showing a null effect. A high heterogeneity was proved again through the obtained results for the *Q*-test and *I*^2^ (*Q* = 1498.94, *p* < 0.01, *I*^2^ = 90.4%). The forest plot is not given due to the lack of an average effect size that is statistically significant in the obtained results.

Mixed: For mixed as a treatment type (meaning that many cultural expressions were included in this type of treatment intervention), *ω* = 0.38 was obtained as an effect size CI [−0.78, 0.02], with *p* = 0.07 proving a null effect. Heterogeneity test results show high heterogeneity (*Q* = 157.48, *p* < 0.01, *I*^2^ = 94.09%) (Figure 2).

Music: Music as treatment type obtained an effect size *ω* = 0.02, CI [−0.10, 0.15] with *p* = 0.70, showing a null effect. In the heterogeneity test, the obtained results proved a high level of heterogeneity (*Q* = 143.55, *p* < 0.01, *I*^2^ = 83.90%) (Figure 3).

Scenic Arts: In the scenic arts intervention treatment type case, the obtained effect size was *ω* = −0.33, CI [−0.52, −0.14] with *p* < 0.01. This concludes that the null hypothesis was rejected, and therefore the effect size was statistically significant. Regarding the heterogeneity test, the obtained results for *Q*-test and *I*^2^ were (*Q* = 46.55, *p* < 0.01, *I*^2^ = 68.03%), proving high heterogeneity (Figure 4).

Visual Arts: The obtained effect size for visual arts as an intervention treatment type was *ω* = −0.01, with CI [−0.28, 0.26], with *p* = 0.95, showing a null effect. High heterogeneity was also shown with the obtained results (*Q* = 440.90, *p* < 0.01, *I*^2^ = 96.75%) (Figure 5).

Reminiscence: There was not enough sample to interpret the results for reminiscence as a treatment type.

Music and Dance: For the music and dance treatment type, there was not enough sample to interpret the results.

#### 3.3.3. Analysis per Evaluation

Cognitive: The effect size obtained when regarding the evaluation of the cognitive function was *ω* = 0.10, CI [0.02, 0.18], with *p* = 0.02, presenting a statistically significant effect size, albeit very low. In the heterogeneity test, it obtained (*Q* = 1622.56, *p* < 0.01, *I*^2^ = 91.13%), proving a high level of heterogeneity. The forest plot is not given due to the lack of an average effect size that is statistically significant in the obtained results.

State of Mind: When considering the evaluation of the state of mind, the obtained effect size was *ω* = −0.45, CI [−0.61, −0.29] with *p* < 0.01, showing an effect size statistically significant. Simultaneously, the heterogeneity test proved a high one (*Q* = 301.13, *p* < 0.01, *I*^2^ = 87.84%) (Figure 6).

General Health State: In relation to the evaluation of the state of general health, the obtained effect size was *ω* = −0.08, CI [−0.23, 0.06] with *p* = 0.25. This shows that the effect size was not statistically significant, concluding that the null hypothesis was not rejected. Regarding the heterogeneity test, the following results (*Q* = 73.64, *p* < 0.01, *I*^2^ = 76.57%) show high heterogeneity (Figure 7).

Quality of Life: The obtained effect size when evaluating the quality of life of the participants of the cultural treatment interventions was *ω* = −0.25, CI [−0.67, 0.17], with *p* = 0.25, showing a null effect. High heterogeneity was also detected through the results obtained for the heterogeneity tests (*Q* = 190.01, *p* < 0.01, *I*^2^=96.48%) (Figure 8).

#### 3.3.4. Analysis per CI Typology

Alzheimer: The obtained effect size when considering Alzheimer as a CI typology was *ω* = 0.05, CI [−0.05, 0.15] with *p* = 0.33, showing a null effect. A high heterogeneity was proved with the obtained results for the heterogeneity tests (*Q* = 253.16, *p* < 0.01, *I*^2^ = 83.92%) (Figure 9).

Dementia: Dementia as a CI typology obtained an effect size of *ω* = −0.30, CI [−0.47, −0.12] with *p* < 0.01. This concludes that the null hypothesis was rejected, and therefore the effect size was statistically significant. Regarding the heterogeneity test, the obtained results showed a high level of heterogeneity (*Q* = 230.93, *p* < 0.01, *I*^2^ = 87.04%) (Figure 10).

Mild Cognitive Impairment: For MCI, CI type *ω* = 0.04 was obtained as effect size CI [−0.28, 0.35], with *p* = 0.81 proving a null effect. The results obtained for the *Q*-test and *I*^2^ demonstrate high heterogeneity (*Q* = 846.85, *p* < 0.01, *I*^2^ = 98.24%) (Figure 11).

Older adults: The obtained effect size when the participants of the cultural treatment interventions were older adults was *ω* = 0.22, CI [−0.01, 0.45], with *p* = 0.07, showing a null effect. High heterogeneity was also shown with the obtained results when testing it (*Q* = 36.42, *p* < 0.01, *I*^2^ = 83.14%) (Figure 12).

Parkinson: The obtained effect size regarding Parkinson as a CI typology was *ω* = −0.04 CI [−0.14, 0.06] with *p* = 0.47, presenting a null effect. In the heterogeneity test, it obtained the following results, showing a high level of heterogeneity: (*Q* = 868.31, *p* < 0.01, *I*^2^ = 87.60%). The forest plot is not given due to the lack of an average effect size that is statistically significant in the obtained results.

#### 3.3.5. Moderator Variables Effects

Examining heterogeneity issues was imperative since the heterogeneity test results were very high for both models. Consequently, a meta-regression model was used to determine how the moderator’s variables could affect the existing heterogeneity.

The overall results obtained by analyzing the effects of moderators’ variables show that not many results reject the null hypothesis. Two statistically significant test results were obtained, both related to the age variable.

The analysis of the relationship between the evaluation type and the results of the meta-analysis reveals the first statistically significant test result. The results, which relate the mean age of the participants and the evaluation of the state of general health with the effect size, must be taken into consideration. With the value *β* = 0.121 (*p* = 0.05), this shows that the younger the group is, the better the results obtained on this evaluation type are, with a bigger effect size.

Another statistically significant result is observed in the analysis of the relationship between the CI typology and the meta-analysis results, as shown in the relevant section. In particular, it appears when the relation between the CI typology and the age and the obtained effect size is analyzed. Having obtained a *β* = 0.221 (*p* = 0.002), it indicates that in older adults CI typology, the younger the participants are, the bigger the effect size becomes. It means that the benefits obtained by the participants with this CI typology are more significant when they are younger.

## 4. Discussion

Twenty-six studies were included in the present MA, with a total of 1201 participants (the sum of the participants taking part in the cited studies). Obtained evidence showed that cultural interventions, when analyzed overall, had no significant effects regarding the benefits they can provide to cognitively impaired participants. Despite the results concerning the effect size, moderators’ variables effects did show that age is a very important variable to consider when estimating the resulting benefits this type of intervention can realistically achieve. 

Nevertheless, the high grade of heterogeneity identified and the results obtained in a previously published SR [22] called for a segmentation of the analyzed data. It is important to mention that in the prior SR, some benefits for participants with CI taking part in cultural interventions were detected.

Multiple partial MAs were conducted, each focusing on studies related to a specific disease, a particular type of treatment, or a specific type of evaluation. This approach allowed for a more nuanced and detailed analysis of the benefits and outcomes associated with cultural interventions within each specific domain. By conducting these partial MAs, the aim was to gain a deeper understanding of the effectiveness and impact of cultural interventions in targeted contexts, providing valuable insights into their outcomes in each of these specific areas. Once partial MAs were conducted, some results supported the effectiveness of cultural interventions as treatment for people living with CI, but some further clarifications are necessary.

The statistically significant result obtained for dementia when conducting a partial MA’ considering the CI typologies demonstrated that cultural interventions can be beneficial in some cases. The obtained results can be related to what the precedent SR concluded [22]. This means that this type of intervention can be beneficial for people living with CI, making their lives more comfortable during the process. However, it will not cure them, make their impairment remit, or stop them from proceeding, as the effect size, despite being statistically significant, is very low. The findings encourage using this type of intervention, but one must be cautious and wary, especially with any expected level of effectiveness and the type of benefits obtained [11,22,58]. 

Scenic or performing arts have been historically described as a useful tool to explore and express the most emotional and affective parts of the human being [59]. The approach this type of intervention can provide to CI patients may be the reason for the statistically significant result, since again, the benefits this type of intervention can provide to the participants are directly related to those outputs related to their emotional wellness more than their clinical well-being [60,61,62]. Again, there is an interrelationship between these meta-analysis results and the ones obtained during the previous SR [22], with the most reported outcomes for performing arts interventions being the ones related to socialization and communication as well as the ones related to emotional well-being. These results reinforce the evidence that cultural interventions have the potential to have a positive impact on non-clinical aspects of CI [58,63,64,65]. 

The state of mind, encompassing emotions, thoughts, and focus, is influenced by various factors and can vary from positive to negative [66,67]. When examining it as an evaluation type, statistically significant results with a substantial effect size were obtained, providing empirical evidence of the proven benefits associated with this evaluation approach. As state of mind is not strictly related to the clinical symptoms of CI suffered by the participants, the obtained evidence was conclusive enough by itself. 

Furthermore, statistically significant results were obtained when analyzing cognitively related evaluations. Nevertheless, in this case, as expected, the results show a very low effect, agreeing with the results obtained in the precedent SR [22]. In that case, overall cognition benefits were cited, but less often than benefits on emotional or sociological aspects. The conclusions we obtained from the previous SR [22] showed that when analyzing improvements in more specific aspects related to cognition, there was not enough evidence. In both cases, there was a low benefit related to the cognition function, and for this reason, it was not convincing enough to emphatically affirm that cultural interventions cause benefits in the cognitive function of participants with CI, and these results should be interpreted with caution.

At this point, and after almost two decades of an increasing number of studies and research trying to show evidence and raise awareness of how arts and culture can be effective in terms of health and well-being [11], and with a significant amount of evidence showing how arts, culture, and health can be potential allies [15,18,58,68] it is necessary to highlight the importance of the results obtained through the current meta-analysis. The obtained results reinforce the idea of how cultural interventions can contribute to cognitively impaired people’s well-being, especially if their heterogeneity is managed properly, turning it into a strength.

The current meta-analysis, as expected, did not reveal huge statistically significant benefits in clinical terms for cognitively impaired people taking part in cultural interventions, and the evidence of their effectiveness in clinical terms is weak. The information that we have gathered raises questions about whether cultural interventions can even be assessed or referred to as treatments. Furthermore, as many of them can be classified as art-therapies, and with the word therapy associated with the act of treating a disease, an injury, or a disability [69], and with enough empiric evidence that disease itself is not being treated through these types of interventions [22,58,63,64,70], it is important to rethink if it is fair to call art-therapies what we understand to be described by this term presently.

It is important to acknowledge certain limitations of this MA. Since our searches were restricted to documents written in English, Catalan, Spanish, and French, it is conceivable that some studies written in other languages may have been overlooked. Furthermore, the MA failed to uncover sufficient studies that included participants with varying typologies of cognitive impairment, especially those not affecting mainly older adults. Certain studies that were included in the MA exhibited inadequate methodological rigor, with deficiencies such as the absence of concealed group allocation and the lack of blinding for participants and those delivering the activity. Moreover, in some cases, certain details were absent regarding the specific approach used to treat/care for the participants. Consequently, the findings should be interpreted with caution.

## 5. Conclusions

In conclusion, our results suggest that cultural interventions, due to their heterogeneity, cannot be assessed as a whole but can highlight the wisdom of performing partial MAs, each focusing on studies pertaining to a specific disease, a particular type of treatment, or a specific type of evaluation, to obtain sharper results, although these results must be interpreted with caution.

When analyzing the gathered data through a partial MA’, the obtained evidence concurred with previous SR conclusions. It shows that cultural interventions can be understood as a tool to obtain mainly benefits regarding the non-clinical aspects of CI, such as state of mind, socialization, self-esteem, or emotional well-being, but not as an intervention to be focused on clinical healing aspects such as the deterioration of cognitive function.

However, far from being unimportant, these results prove categorically that these types of interventions provide benefits in terms of well-being and socialization. Above all, considering that most causes of CI still have no cure today, these interventions can help people with cognitive impairment to have a better quality of life in emotional and social aspects and to live this stage of life in a more pleasant way.

## Figures and Tables

**Figure 1 healthcare-11-01854-f001:**
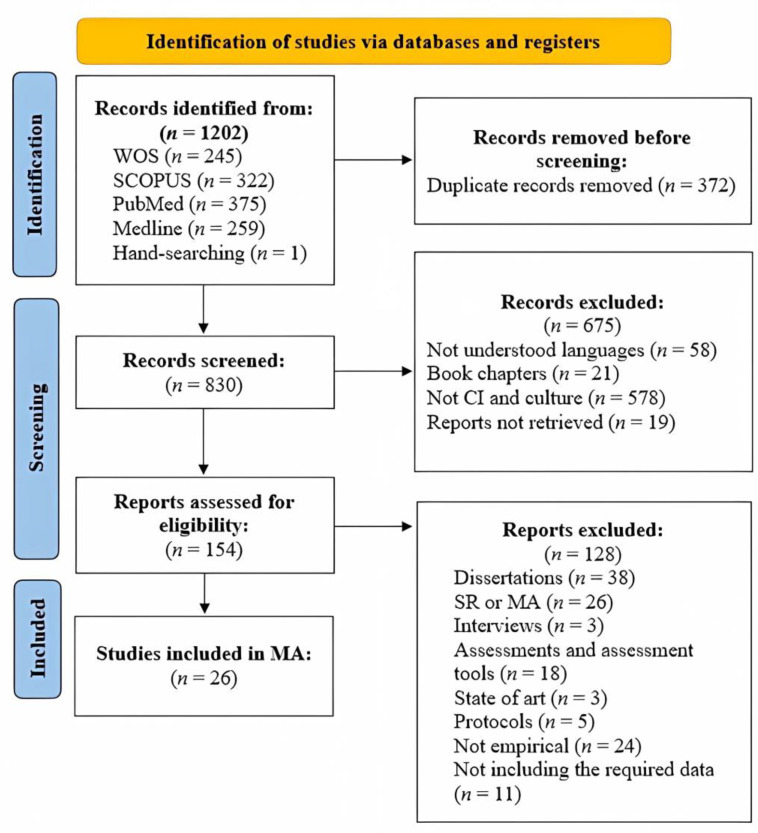
PRISMA flow chart of included papers.

**Figure 2 healthcare-11-01854-f002:**
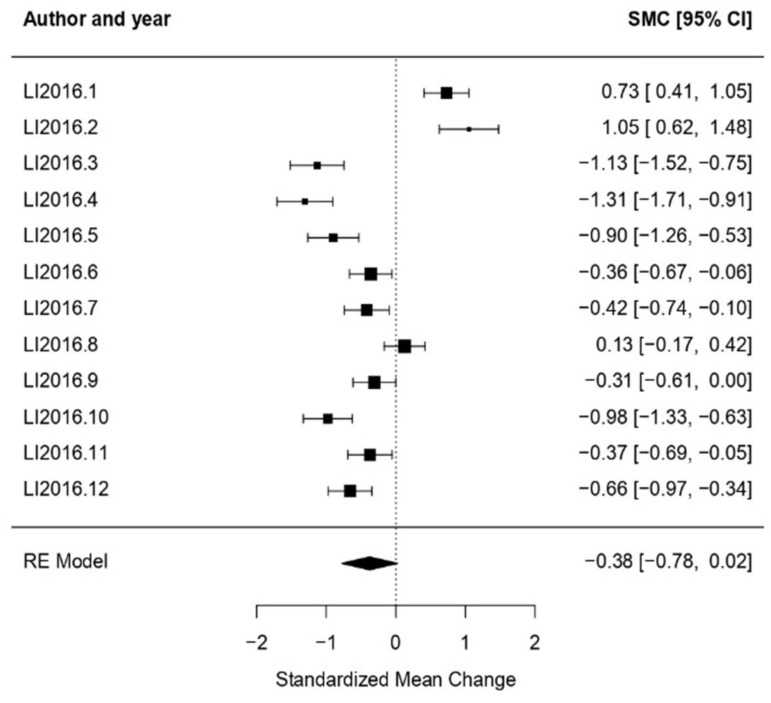
Forest plot of partial meta-analysis. Analysis per treatment—Mixed [39].

**Figure 3 healthcare-11-01854-f003:**
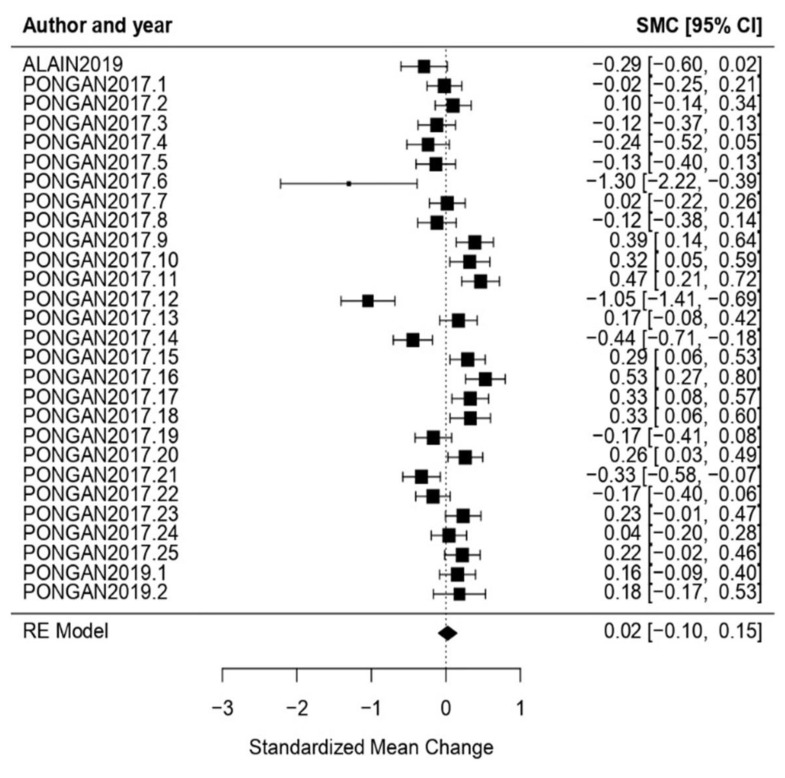
Forest plot of partial meta-analysis. Analysis per treatment—Music [44,45,54].

**Figure 4 healthcare-11-01854-f004:**
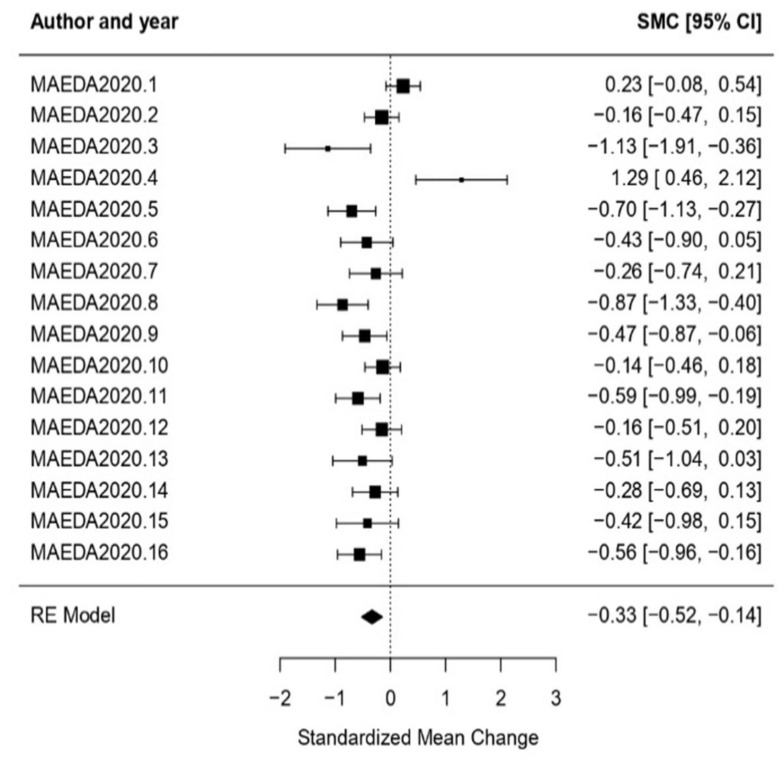
Forest plot of partial meta-analysis. Analysis per treatment—Scenic Arts [51].

**Figure 5 healthcare-11-01854-f005:**
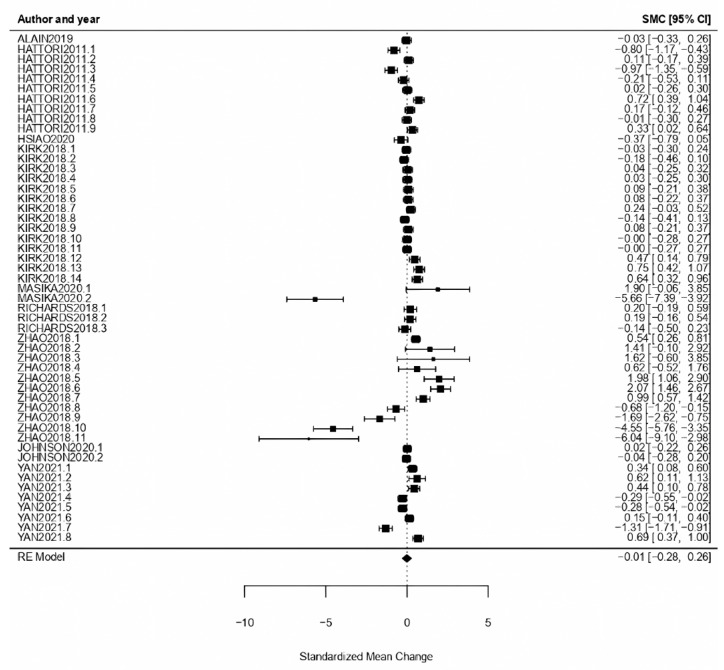
Forest plot of partial meta-analysis. Analysis per treatment—Visual Arts [37,40,41,42,43,50,52,54,57].

**Figure 6 healthcare-11-01854-f006:**
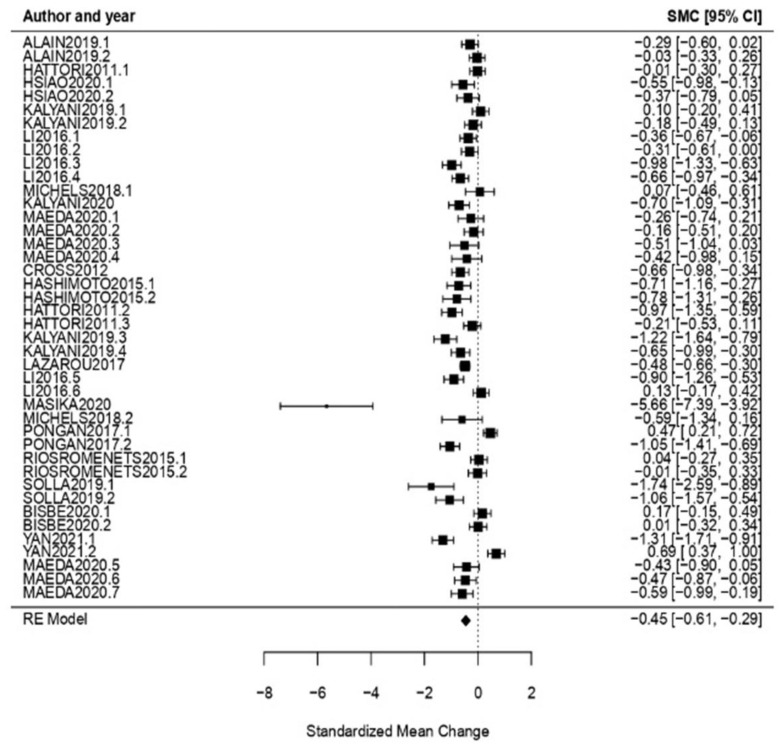
Forest plot of partial meta-analysis. Analysis per evaluation—State of Mind [10,34,35,38,39,41,42,44,46,48,49,50,51,52,53,54,56].

**Figure 7 healthcare-11-01854-f007:**
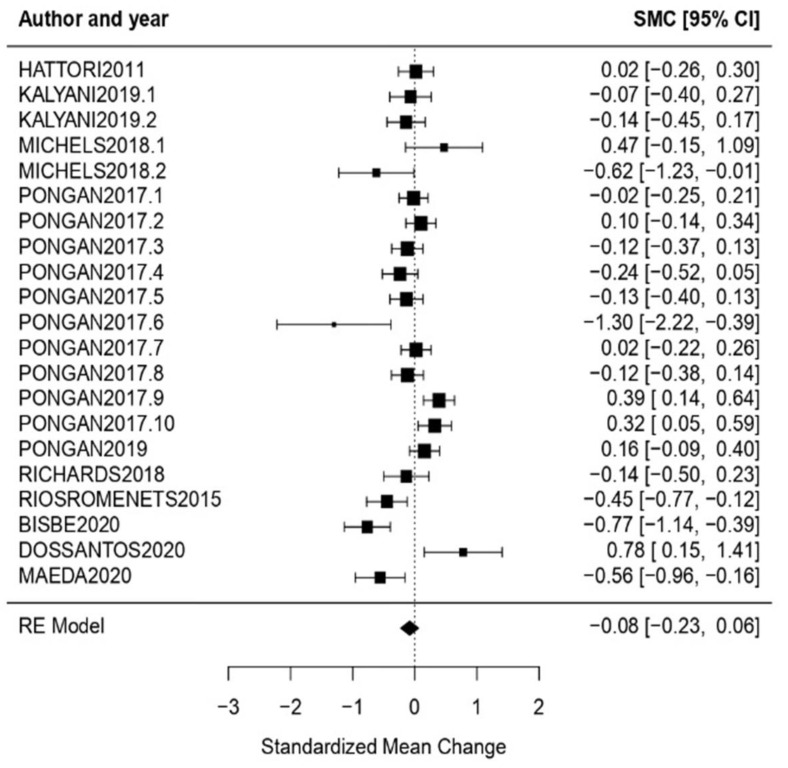
Forest plot of partial meta-analysis. Analysis per evaluation—General Health State [34,36,38,44,45,50,51,53,56,57].

**Figure 8 healthcare-11-01854-f008:**
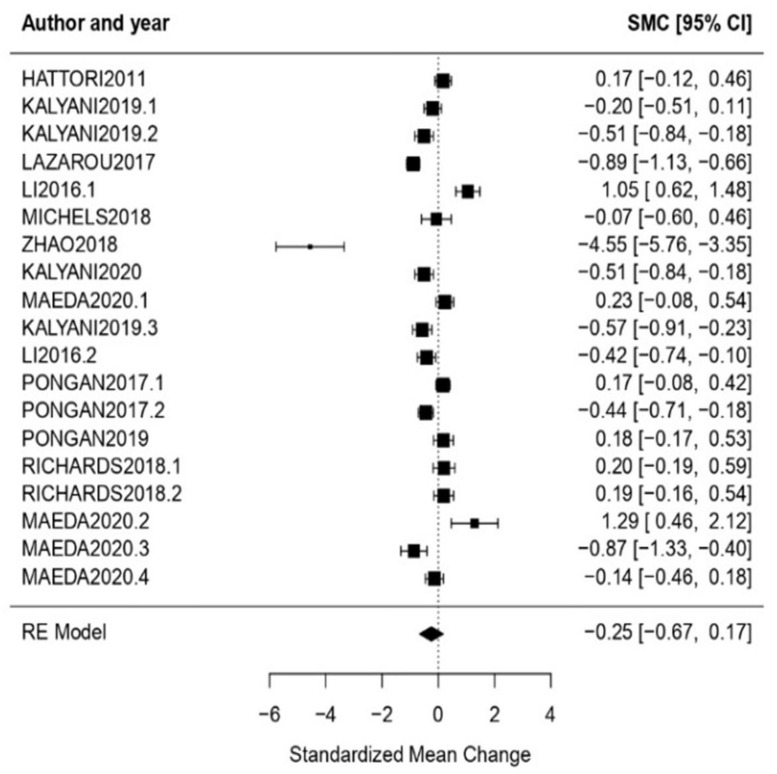
Forest plot of partial meta-analysis. Analysis per evaluation—Quality of Life [34,35,39,40,44,45,46,50,51,56,57].

**Figure 9 healthcare-11-01854-f009:**
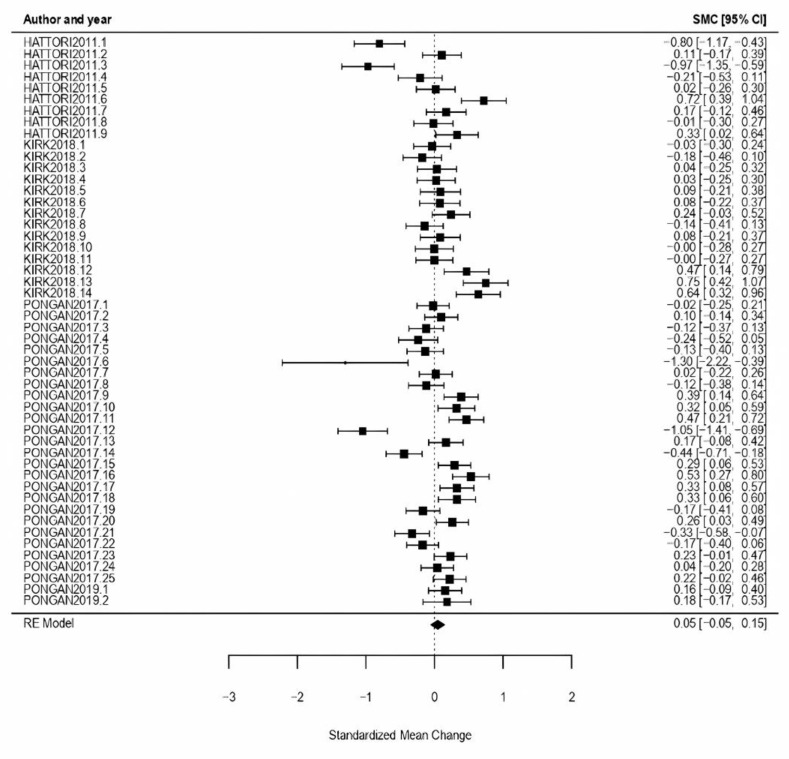
Forest plot of partial meta-analysis. Analysis per CI typology—Alzheimer [43,44,45,50].

**Figure 10 healthcare-11-01854-f010:**
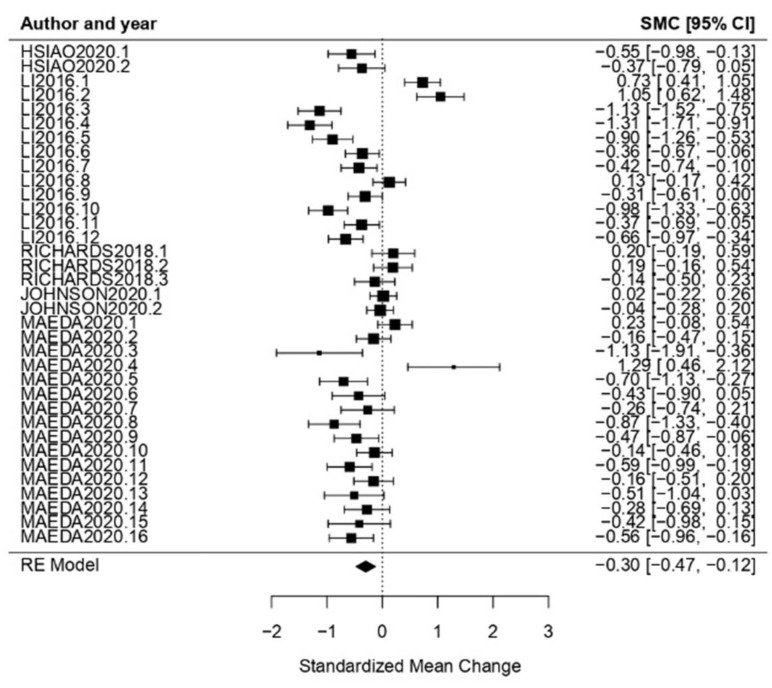
Forest plot of partial meta-analysis. Analysis per CI typology—Dementia [37,39,51,52,57].

**Figure 11 healthcare-11-01854-f011:**
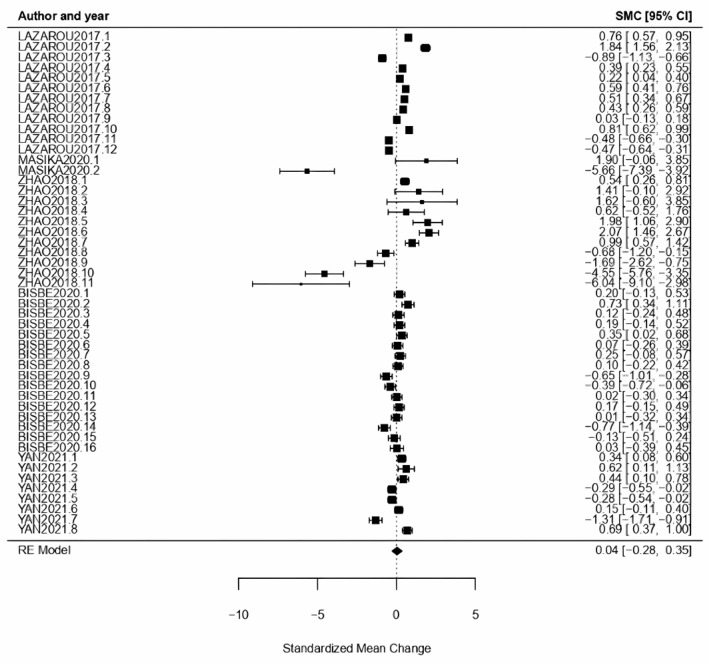
Forest plot of partial meta-analysis. Analysis per CI typology—Mild Cognitive Impairment [40,41,42,46,53].

**Figure 12 healthcare-11-01854-f012:**
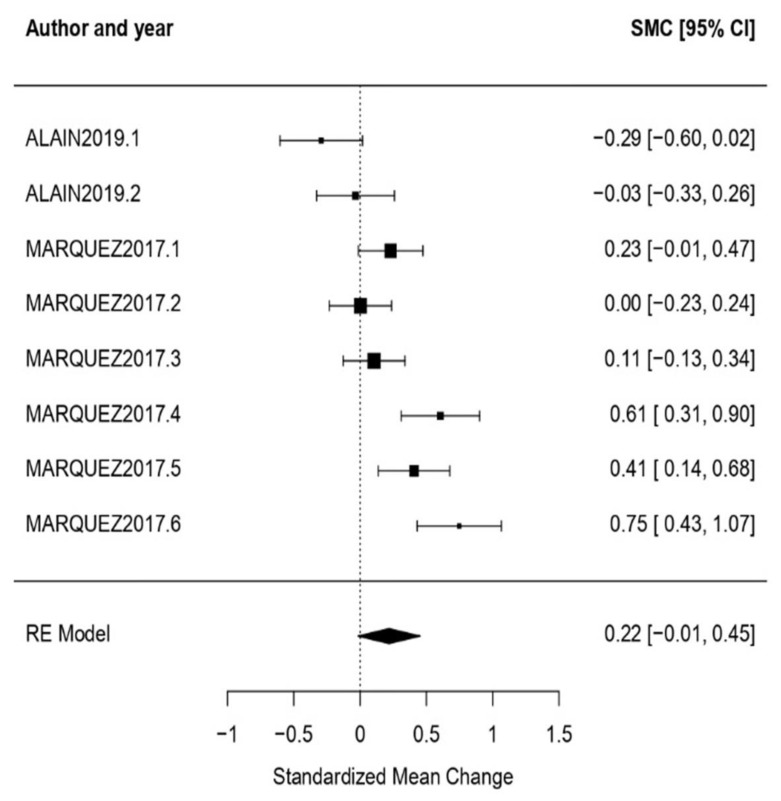
Forest plot of partial meta-analysis. Analysis per CI typology—Older adults [54,55].

**Table 1 healthcare-11-01854-t001:** Included studies characteristics.

Reference	Year	Country	CI Typology	Nº CI Participants	Nº Groups	Mean Age Treatment	Mean Age Control	Treatment Type	Treatment Design	Study Design
[54]	2019	United States	Older adults	53	3	67.70	68.50	Group-Music	Cultural treatment vs. Cultural treatment vs. Control	Quasi-randomized
[53]	2020	Spain	Mild cognitive impairment	31	2	72.88	77.29	Group-Dance	Cultural treatment vs. Treatment	Randomized
[10]	2012	United States	Cognitive impaired	100	2	75.58	77.30	Group-Music and dance	Cultural treatment vs. Cultural treatment	Randomized
[47]	2016	Italy	Parkinson	16	2	66.00	70.00	Group-Dance	Cultural treatment vs. Treatment	Randomized
[36]	2020	Brazil	Parkinson	18	2	68.60	64.20	Group-Dance	Cultural treatment vs. Treatment	Not randomized
[49]	2015	Japan	Parkinson	46	3	67.90	69.70	Group-Dance	Cultural treatment vs. Treatment	Quasi-randomized
[50]	2011	Japan	Alzheimer	39	2	75.30	73.30	Group-Visual art	Cultural treatment vs. Treatment	Randomized
[52]	2020	Republic of China	Dementia	54	3	/	/	Group-Reminiscence	Cultural treatment vs. Treatment vs. Comparison	Randomized
[37]	2020	Canada	Dementia	53	2	79.70	82.30	Group-Visual art	Cultural treatment vs. Control	Randomized
[34]	2019	Australia	Parkinson	33	2	65.24	66.50	Group-Dance	Cultural treatment vs. Control	Not randomized
[35]	2020	Australia	Parkinson	33	2	65.24	66.50	Group-Dance	Cultural treatment vs. Control	Quasi-randomized
[43]	2018	Denmark	Alzheimer	43	2	80.18	79.86	Group-Visual art	Cultural treatment vs. Control	Randomized
[46]	2017	Greece	Mild cognitive impairment	129	2	65.89	67.92	Group-Dance	Cultural treatment vs. Control	Randomized
[39]	2016	China	Dementia	48	2	83.10	81.80	Group-Mixed	Cultural treatment vs. Control	Quasi-randomized
[51]	2020	Japan	Dementia	34	2	84.00	82.00	Group-Scenic arts	Cultural treatment vs. Control	Randomized
[55]	2017	United States	Older adults	57	2	64.80	66.40	Group-Dance	Cultural treatment vs. Control	Randomized
[42]	2020	China	Mild cognitive impairment	39	2	73.40	72.00	Group-Visual art	Cultural treatment vs. Treatment	Randomized
[56]	2018	United States	Parkinson	13	2	66.44	75.50	Group-Dance	Cultural treatment vs. Control	Randomized
[44]	2017	France	Alzheimer	59	2	78.80	80.20	Group-Music	Cultural treatment vs. Cultural treatment	Randomized
[45]	2019	France	Alzheimer	59	2	78.80	80.20	Group-Music	Cultural treatment vs. Cultural treatment	Randomized
[57]	2018	United States	Dementia	26	4	74.80	74.00	Group-Visual art	Cultural treatment vs. Control	Randomized
[38]	2015	Canada	Parkinson	33	2	63.20	64.30	Group-Dance	Cultural treatment vs. Treatment	Randomized
[48]	2019	Italy	Parkinson	20	2	67.80	67.10	Group-Dance	Cultural treatment vs. Control	Randomized
[7]	2013	Italy	Parkinson	24	2	61.60	65.00	Group-Dance	Cultural treatment vs. Treatment	Randomized
[41]	2021	China	Mild cognitive impairment	48	2	69.00	66.00	Group-Visual art	Cultural treatment vs. Control	Randomized
[40]	2018	China	Mild cognitive impairment	93	2	70.60	69.50	Group-Visual art	Cultural treatment vs. Control	Randomized

## Data Availability

The data that support the findings of this study are available from the corresponding author, [L.D.-L.], upon reasonable request.

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
