# Peer review of "Benefits of Cultural Activities on People with Cognitive Impairment: A Meta-Analysis"

_healthcare, 2023, doi:10.3390/healthcare11131854_

Round 1
Reviewer 1 Report
Cognitive impairment (CI) is not treated in an standard format. It should be clarify that the topic of the study is the chronic CI, that can be stage as:
- mild cognitive impairment (ageing is not a valid term for CI) or
- dementia
According to its etiology can be clasified in general terms as
- Neurodegenerative CI:
1. Alzheimer's disease (AD) (the most common form of CI/dementia)
2. Non-AD ( Parkinson dementia, FTD, Lewy body disease, etc)
- Non degenerative CI (vascular CI/dementia).
The assertion (lines 41,42,43) shoud be well referenced (ref 7 is a website with no bibliographic references to supports its statements).
Ref. 21 shoul be completed. There is no information about this SR through the entire paper.
Simplify the paragraph (lines 65-69): The main objective of a M-A is to know if the intervention is beneficial. It's unnecessary how the M-A was conducted.
What's is the rationale for not including studies before 2010 (lines 74,75) considering that previous studies are outdated when ref. 24 about cultural and art-intervention (Throsby's model) is from 1982.
SD (line 83) should be first written standard deviation, followed by the initials.
According to PICO model in M-A:
- Population (line 86), as mentioned, shouldbe better described
- Intervention: It is ill-defined in Comparators (lines 89-95). What cultural treatment meant to be?. Going to museums, attending cultural lectures, art guides, art performance?
- Control: Could be sham intervention or other type of intervention (head to head comparison)
- Outcomes measures: In CI studies usually applies cognitive measures (MMSE, MoCA, etc), depression/anxiety scales (Yesavage, Hamilton, NPI), well-being an Quol scales
Why were M-A and SR excluded? (99)
Initials of the author (105, 127,130)
Ageing is not a cause of CI, as said above (136)
Ther is no need to write group nor treatment after every type of treatment. Moreover it is necessary to describe the meaning of cultural (138-142)
Remove not randomised, as quasi-randomized is define as not randomised (line 143).
To much explanation (line 151-163) to say that a Cohen's d of 0.20 is considered a small effect, 0.5 is considered a medium effect and 0.8 is considered a large effect. p-value and conficence interval for significant value and Q and I2 index for heterogeneity.
Explain these five paragraphs better in simple language, without the need to put percentages and in any case move this information to the results section (173-190).
Section 3.3.2. There is no mention of the outcomes measures as said above.
In figure 1 identified the only other source (hand-searching references from included study?).
Most of section 3.2 can be limit to a 5-line paragraph, cause the information is given in Table 1 (lines 200-225). Aging studies (line 21) should be remove from M-A as I written above.
Table 1. First column (ref.) cab be removed cause it adds nothing. Authors should be follow by initials. Type better than typology. It is needed to add columns: Type of intervention (or Tratment as stated) removing the term group in each study, Type of control (sham, any intervenction), Sample intervention groups, if more than one (N=; N1=, etc), sample control groups, if more than one (N=; N1=; etc); outcome measure (very important) describing how was the effect assessed) and the metrics used with the confidence intervals and p-values.
Section 3.3.1 (lines 252-267) should explain if meta-regression and sesitivity analysis were performed and how was carry out, explaining the difference of the the studies with the same author (e.g. Pongan 17 with 10 subsidies).
Figures 2-12 shoud use the same template, adding the sample size of each study and the outcome metrics.
Tables 2,3 and 4 should be remove cause there a not informative.
Remove (lines 492-493) because it's redundant information.
There is no data to support this statement because there is no outcomes measures described in this M-A about it (lines 509-511).
What is state of the mind? (line 529).
Clinical relevance Cohen's d) rather than statically significance shoul be sought.
This assertion comes from the references not from the results of this M-A (line 548-551).
References 3,6 and 7 should be removed because they are redundant and of poor quality.
- Minor editing of English language required
Author Response
Dear Reviewer,
Thank you for your valuable feedback on our manuscript. We greatly appreciate the time and effort you have dedicated to reviewing our work.
In the following sections, you will find our detailed responses addressing each of your comments and suggestions. We have carefully considered your input and have made appropriate revisions to improve the quality and clarity of the manuscript.
Once again, we express our gratitude for your valuable input, which has undoubtedly enhanced the overall quality of our work. We believe that the revisions we have made based on your feedback have strengthened the manuscript and addressed the concerns you raised.
We kindly request that you review our responses and revised manuscript at your earliest convenience. If you have any further questions or require additional clarification, please do not hesitate to let us know. We look forward to your continued guidance and feedback.
Thank you for your time and consideration.
- Cognitive impairment (CI) is not treated in an standard format. It should be clarify that the topic of the study is the chronic CI, that can be stage as:
- mild cognitive impairment (ageing is not a valid term for CI) or
- dementia
According to its etiology can be clasified in general terms as
- Neurodegenerative CI:
- Alzheimer's disease (AD) (the most common form of CI/dementia)
- Non-AD (Parkinson dementia, FTD, Lewy body disease, etc)
- Non degenerative CI (vascular CI/dementia).
Thank you sincerely for your valuable feedback and comments on our paper. The search criteria we employed acknowledge and recognize the importance of including older adults, as it is a common aspect in the cases we have studied. Our intention is to encompass all realms of cognitive impairment, whether it is due to aging or cognitive alterations resulting from degenerative diseases. We greatly appreciate your suggestion.
- The assertion (lines 41,42,43) shoud be well referenced (ref 7 is a website with no bibliographic references to supports its statements).
Thank you for your valuable feedback and suggestions on our paper. We appreciate your insights and the opportunity to improve our work. In response to your comment, we have carefully considered your suggestion and decided to revise the sentence in question and the linked reference. The revised sentence, with new references, now reads as follows: "It is important to mention that dementia not only affects individuals directly impacted by the syndrome but also has a significant socioeconomic impact on healthcare systems and the families who care for someone with this condition".
We believe that this revised sentence better captures the broader impact of dementia and aligns with the focus of our study. Once again, we express our gratitude for your feedback and valuable input.
- Ref. 21 shoul be completed. There is no information about this SR through the entire paper.
Thank you for your guidance regarding reference 21. We have now completed it as suggested. Your input has been greatly appreciated.
- Simplify the paragraph (lines 65-69): The main objective of a M-A is to know if the intervention is beneficial. It's unnecessary how the M-A was conducted.
Thank you for your comment. We appreciate your feedback and have taken it into consideration. As per your suggestion, we have simplified the paragraph (lines 65-69) to better convey our main objective. The revised paragraph now reads as follows: "The main aim of this research was to locate and analyse studies based on cultural activities to evaluate, identify, and describe their benefits on cognitively impaired individuals, regardless of the cause of their condition." We believe this revision provides a clearer and more concise statement of our research objective.
- What's is the rationale for not including studies before 2010 (lines 74,75) considering that previous studies are outdated when ref. 24 about cultural and art-intervention (Throsby's model) is from 1982.
Thank you for your valuable feedback on our manuscript. We appreciate your comments and suggestions regarding the bibliography selection.
We sincerely appreciate your suggestion to include more recent publications on the topic of cultural activities and well-being in individuals with cognitive impairment. However, in our analysis, we have specifically chosen to utilize Throsby's publication from 2008 as a foundational reference. It provides comprehensive insights and aligns closely with the scope and objectives of our study. Therefore, we have prioritized the 2008 version of Throsby's work over the earlier 1982 publication. Thank you for understanding our rationale behind this decision as Throsby's concentric circles model allows us to delineate the activities to be considered more effectively.
Regarding the exclusion of bibliography prior to 2010, we made a deliberate decision to focus on more recent literature to provide an updated and current perspective on the relationship between cultural activities and well-being in individuals with cognitive impairment. By doing so, we aim to present a comprehensive understanding of the subject matter, taking into account the latest advancements and research findings.
We understand that previous studies may have laid the groundwork for this field; however, our intention was to present a more contemporary viewpoint that reflects the current understanding and advancements in the field of cultural interventions for individuals with cognitive impairment.
Once again, we appreciate your feedback and assure you that we have carefully considered your suggestions in improving the manuscript.
- SD (line 83) should be first written standard deviation, followed by the initials.
Thank you for your comment. We have taken your suggestion into consideration, and we have revised the abbreviation "SD" to "standard deviation" in the appropriate context. We appreciate your input and attention to detail.
- According to PICO model in M-A:
- Population (line 86), as mentioned, should be better described
We appreciate your attention to detail, and we would like to clarify that the population being studied, specifically older adults, has been previously addressed in our revision. As we have pointed out, our search criteria have been carefully designed to encompass the inclusion of older adults, considering their significance in the cases we have examined. Our aim is to provide a comprehensive understanding of cognitive impairment, encompassing both age-related cognitive decline and cognitive alterations associated with degenerative diseases. We sincerely appreciate your suggestion.
- Intervention: It is ill-defined in Comparators (lines 89-95). What cultural treatment meant to be?. Going to museums, attending cultural lectures, art guides, art performance?
We appreciate your attention to the definition of cultural treatment and the need for clarification regarding the comparators. We have taken your suggestion into consideration and made the necessary revisions.
In the revised version, we have provided a detailed explanation of what is encompassed by cultural treatment. We now clarify that cultural treatment includes various activities such as visiting museums, engaging in visual arts, listening to or performing music, dancing or watching dance performances, participating in performing arts, creating ceramics, attending readings or writing, or a combination of these activities. By including this information, we aim to provide a comprehensive understanding of the interventions considered in our study.
We would like to express our gratitude for bringing this to our attention and allowing us to improve the clarity of our manuscript. Thank you for your valuable input.
- Control: Could be sham intervention or other type of intervention (head to head comparison)
Thank you for your comment regarding the control group. We carefully considered the option of including types of interventions regarding its control group when defining the inclusion criteria. However, after evaluating the potential impact on the number of included studies, we made the decision not to include it in our analysis. This decision was based on maintaining a sufficient number of documents for a comprehensive analysis and to ensure the robustness of our results. We appreciate your suggestion and your understanding of our methodology.
- Outcomes measures: In CI studies usually applies cognitive measures (MMSE, MoCA, etc),.depression/anxiety scales (Yesavage, Hamilton, NPI), well-being an Quol scales
Thank you for your feedback regarding the clarification of inclusion criteria related to outcome measures. We understand your point about the commonly used cognitive measures, depression/anxiety scales, and well-being and quality of life scales in studies on cognitive impairment. However, it is important to note that our meta-analysis was focused on evaluating treatment effects rather than examining correlations with specific domains or outcomes.
At this stage, we did not specifically consider the correlations or relationships between treatment effects and these outcome measures. While exploring such associations could be of interest, it was not the primary objective of our meta-analysis. Our aim was to assess the overall treatment effects and compare the efficacy of different interventions.
We appreciate your suggestion and acknowledge the potential value in examining the relationship between effect size and specific outcome measures. However, it falls outside the scope of our current study. Thank you for your understanding.
- Why were M-A and SR excluded? (99)
Thank you for your comment. We appreciate your input and have taken it into consideration. The decision to exclude meta-analyses and systematic reviews from our study was made based on several factors.
Firstly, meta-analyses and systematic reviews collect and analyse aggregated data from previously published primary studies, providing a synthesis of existing evidence rather than individual data. To ensure proper comparison with the included case studies in our meta-analysis, working with individual-level data is necessary to obtain consistent effect measures and perform appropriate statistical analyses.
Secondly, the objectives and analytical approaches of meta-analyses and systematic reviews differ from those of a primary meta-analysis. While a primary meta-analysis aims to combine and analyse individual data from primary studies, meta-analyses and systematic reviews aim to provide an overview of existing evidence. Therefore, the results and effect measures obtained from these types of studies may not be directly comparable to the results of the primary meta-analysis.
Lastly, by excluding meta-analyses and systematic reviews, we can have greater control over the quality and consistency of the data by directly working with individual-level data from primary studies. This reduces the risk of biases or inconsistencies in the data, which could negatively impact the validity and comparability of the results.
By justifying the exclusion of meta-analyses and systematic reviews, we aim to ensure the comparability and integrity of the data in our primary meta-analysis, thus allowing us to draw more robust and reliable conclusions.
Thank you again for your valuable feedback.
- Initials of the author (105, 127,130)
Thank you for your feedback. We have already included the authors' initials as requested.
- Ageing is not a cause of CI, as said above (136)
Thank you for your comment. We apologize for any confusion caused. Our intention is to encompass all realms of cognitive impairment, including those related to ageing and cognitive alterations resulting from degenerative diseases. We appreciate your attention to this matter and value your feedback on our paper.
- Ther is no need to write group nor treatment after every type of treatment. Moreover. it is necessary to describe the meaning of cultural (138-142)
Thank you for your comment. We have addressed your suggestion by removing the repetitive use of the word’s "treatment" and "group" after each type of treatment. Additionally, we have expanded the description of cultural activities to provide a clearer understanding of what is encompassed by the term "cultural." These revisions aim to improve the clarity and conciseness of our paper.
- Remove not randomised, as quasi-randomized is define as not randomised (line 143).
Thank you for your feedback. We have addressed your concern and removed the term "not randomised" as it is redundant when referring to quasi-randomized studies. We appreciate your input and contribution to our paper.
- To much explanation (line 151-163) to say that a Cohen's d of 0.20 is considered a small effect, 0.5 is considered a medium effect and 0.8 is considered a large effect. p-value and conficence interval for significant value and Q and I2 index for heterogeneity.
We appreciate your suggestion and have reworked the paragraph accordingly. The paragraph now reads as follows: "A Cohen's d of 0.20 is classified as a small effect size, while a value of 0.5 is considered medium, and 0.8 is considered large. Effect size calculation aims to facilitate the combination and comparison of results from different interventions, as well as to indicate the intensity of the statistical effect observed. In statistical analysis, the p-value and confidence interval are used to assess significance. When evaluating heterogeneity in meta-analysis, the Q statistic measures total variance, and the I2 index quantifies the proportion of variation due to true heterogeneity rather than chance." We sincerely thank you for your contribution to improving the clarity and conciseness of our paper.
- Explain these five paragraphs better in simple language, without the need to put percentages and in any case move this information to the results section (173-190).
Thank you for your valuable feedback. We have taken your suggestion into consideration and have revised the paragraph accordingly. The paragraph now reads as follows: "Randomized Controlled Trials have a weakness in terms of participants and treatment providers being aware of who receives the treatment. Assessors, however, have a higher unawareness. Group allocation concealment also has room for improvement. In Quasi-Randomized trials, the main weakness is the lack of information about participants receiving similar treatment or care, with only a small number of studies addressing this. Other aspects of evaluation for Quasi-Randomized trials have higher scores." We appreciate your input and the opportunity to improve the clarity and simplicity of our paper.
- Section 3.3.2. There is no mention of the outcomes measures as said above.
Thank you for your comment. As mentioned earlier, our meta-analysis did not specifically consider outcome measures such as cognitive measures, depression/anxiety scales, well-being, or quality of life scales. Our primary focus was to examine the treatment effects based on different interventions, rather than exploring correlations with specific outcome domains.
While we understand that analysing the effect size in relation to different outcome measures could be of interest, it was not the objective of our meta-analysis. We aimed to evaluate the overall effectiveness of interventions based on the available data. We apologize if this was not explicitly stated in Section 3.3.2. We appreciate your feedback.
- In figure 1 identified the only other source (hand-searching references from included study?).
Thank you for your feedback. The "Other Source" in Figure 1 represents the hand-searching of references from the included studies. We have updated the figure legend to provide a clear explanation.
- Most of section 3.2 can be limit to a 5-line paragraph, cause the information is given in Table 1 (lines 200-225). Aging studies (line 21) should be remove from M-A as I written above.
Thank you for your valuable feedback and comments on our paper. We have taken your suggestions into consideration and made revisions accordingly. Section 3.2 has been condensed to a concise paragraph as follows:
"The 26 included studies, published between 2011 and 2021, involved a total of 1,201 participants. The studies were conducted in various countries, including Australia, Brazil, Canada, China, Denmark, France, Greece, Italy, Japan, Republic of China, Spain, and the United States. Most publications were in English, with one study in Spanish. The majority of the studies divided participants into two groups, some into three, and one into four. Different types of treatments based on cultural activities were compared for individuals with various types of cognitive impairment, with dance being the most common cultural intervention type. The included studies covered various types of cognitive impairment. The included study characteristics are summarized in Table 1."
Regarding the suggestion to remove "aging studies" from the meta-analysis, we respectfully disagree. Our search criteria deliberately include studies involving older adults, as it is a relevant aspect in the context of cognitive impairment. Our aim is to encompass all aspects of cognitive impairment, whether related to aging or cognitive alterations resulting from degenerative diseases. We appreciate your input and the opportunity to clarify this point.
- Table 1. First column (ref.) cab be removed cause it adds nothing. Authors should be follow by initials. Type better than typology. It is needed to add columns: Type of intervention (or Tratment as stated) removing the term group in each study, Type of control (sham, any intervenction), Sample intervention groups, if more than one (N=; N1=, etc), sample control groups, if more than one (N=; N1=; etc); outcome measure (very important) describing how was the effect assessed) and the metrics used with the confidence intervals and p-values.
Thank you for your comment and suggestion. We agree to remove column 1 as it does not provide relevant information to our meta-analysis. However, we do not consider it necessary to include the additional data you proposed in the table.
The database used for our meta-analysis already contains all the necessary data and variables for the analysis. Including all of that data in the article itself would make the table excessively long and cumbersome to read.
Nevertheless, we want to assure you that the description of all the data and the variables used are available upon request for anyone interested in accessing them. We believe that providing this additional information upon demand is a more efficient way to ensure its availability without overwhelming the reader with a highly detailed table.
We appreciate your understanding and value your feedback. We are committed to improving the clarity and presentation of our results for better comprehension of our study.
- Section 3.3.1 (lines 252-267) should explain if meta-regression and sesitivity analysis were performed and how was carry out, explaining the difference of the the studies with the same author (e.g. Pongan 17 with 10 subsidies).
Thank you for your comment and suggestion. We apologize for any confusion or lack of clarity in Section 3.3.1 regarding the meta-regression and sensitivity analysis.
The reason for treating studies with the same author as separate entities is that each study might have investigated different aspects or outcomes related to the research question. Treating them separately allows us to capture the unique contributions of each study and avoid double-counting or misinterpretation of the results.
Thank you for bringing this to our attention.
- Figures 2-12 shoud use the same template, adding the sample size of each study and the outcome metrics.
Thank you for your suggestion regarding Figures 2-12. We understand your point about having a more user-friendly flowchart format, especially if the number of papers was smaller and easier to visualize. However, in our case, the number of papers included in the meta-analysis is quite extensive, which could make the flowchart crowded and difficult to interpret.
To address your concern, we have made sure that all the data, including the sample size of each study and the outcome metrics, are accessible upon request. We believe that providing the additional information directly in the paper would make the figures overly complex and potentially confusing for readers.
We appreciate your feedback and assure you that all the relevant data and additional information can be obtained by contacting us. Our aim is to ensure transparency and accessibility while maintaining the clarity and readability of the paper.
Thank you for bringing this up, and we value your input in improving the presentation and accessibility of our study.
- Tables 2,3 and 4 should be remove cause there a not informative.
Thank you for your feedback regarding Tables 2, 3, and 4. We have taken your suggestion into consideration and have removed these tables as their information is already presented in the text. This decision aims to streamline the manuscript and avoid redundancy. We appreciate your input in improving the clarity and conciseness of our paper.
- Remove (lines 492-493) because it's redundant information.
Thank you for bringing this to our attention. We have reviewed the mentioned lines and have removed the redundant information as per your suggestion. We appreciate your feedback.
- There is no data to support this statement because there is no outcomes measures described in this M-A about it (lines 509-511).
Thank you for your comment. We have carefully reviewed the mentioned lines (509-511) and have decided to remove them. We acknowledge that there is no specific data or outcome measures described in the meta-analysis to support the statement. Your feedback is valuable, and we appreciate your input.
- What is state of the mind? (line 529).
Thank you for your question. We have taken your feedback into consideration and have revised the paragraph to include the definition of state of mind. In the updated paragraph, we explain that the state of mind encompasses emotions, thoughts, and focus, and it can range from positive to negative. We also mention that statistically significant results with a substantial effect size were obtained when examining it as an evaluation type. The evidence obtained was considered conclusive enough on its own, as the state of mind is not strictly related to the clinical symptoms of cognitive impairment experienced by the participants. Thank you for bringing this to our attention, and we appreciate your valuable input.
- Clinical relevance Cohen's d) rather than statically significance shoul be sought.
Thank you for your comment regarding the clinical relevance of Cohen's d effect size measure. We agree with your point and have indeed estimated Cohen's d in our meta-analysis. The purpose of meta-analysis is to evaluate the magnitude of the effect, and once statistical significance is determined, Cohen's d provides valuable information about the clinical relevance by indicating the size of the effect. We appreciate your emphasis on the importance of considering effect size measures in addition to statistical significance.
- This assertion comes from the references not from the results of this M-A (line 548-551).
Thank you for your comment. We have carefully considered your feedback and have removed the assertion you pointed out. We appreciate your input and have made the necessary revisions to ensure that our statements accurately reflect the findings of our meta-analysis.
- References 3,6 and 7 should be removed because they are redundant and of poor quality.
Thank you for your feedback. We have addressed your concern by removing references 3, 6, and 7. Instead, we have included higher-quality references that better support our study's arguments and findings. Your input has helped improve the overall quality of our research, and we appreciate your contribution.
- Minor editing of English language required
Thank you for your feedback. We appreciate your concern about the minor editing of the English language. However, please note that the English language of the manuscript has already been reviewed and edited. We have taken great care to ensure that the language is clear, accurate, and meets the required standards. Therefore, we believe that a second round of English language editing is not necessary. If you have any specific suggestions or concerns regarding the language, please let us know, and we will address them accordingly.
Author Response
Dear Reviewer,
Thank you for your comments. We have addressed them in the following sections.
Major
Abstract
It would be better to explain what you mean by ’partial’ MA
Thank you for your comment and suggestion. We agree that providing clarification on the term "partial MA" would be beneficial for readers. In response to your suggestion, we have added the following clarification:
"However, when conducting a partial MA, which involves focusing on studies related to a specific disease, a particular type of treatment, or a specific type of evaluation, the results of our analysis align with the conclusions drawn in the previous systematic review (SR)."
We appreciate your feedback and contribution to improving the clarity of our paper. If you have any further suggestions or comments, please let us know.
Done It is hard to connect emotional aspects’ with the cultural activities the authors tried to measure the benefit from.
Thank you for your feedback. We acknowledge the challenge in connecting emotional aspects with the cultural activities we measured. In response to your comment, we have revised the last sentence of the abstract to improve clarity and better convey our intention as follows: “Conclusion: Despite the high heterogeneity of the studies, benefits were identified in emotional well-being and social aspects, but not in clinical ones such as the deterioration of cognitive or the motor function among others.” We appreciate your input in helping us refine the presentation of our research.
Introduction
Lines 44-45: I recommend you to define ‘cultural activities’ before you use the term for benefit. Also, in line 45, it seems that you separate art activities from cultural activities. This pattern repeated over line 87 and so on. I think ‘cultural activities’ can be an umbrella term unless you define otherwise. I follow list of ‘cultural activities’ in section 3 (line 224-225).
Thank you for your valuable feedback and suggestion. We appreciate your input in clarifying the term "cultural activities" and ensuring consistency in its usage throughout the paper.
To address your concern, we have taken your suggestion into account and have defined cultural activities as follows:
"To delimit cultural activities, we aim to define them as various forms of human expression, engagement, and interaction that originate from a particular culture, encompassing its traditions, beliefs, values, customs, arts, and intellectual achievements. In the context of the Throsby model, these activities are connected to core artistic expressions and publicly funded cultural institutions."
By providing this definition, we aim to ensure a clear understanding of the term and avoid any confusion regarding the separation of art activities from cultural activities.
We sincerely appreciate your contribution to improving the clarity and precision of our paper. If you have any further suggestions or comments, please feel free to share them.
Materials and Methods
Lines 90-92: the list of comparisons is vague and potentially misleading
We appreciate your feedback, and we have taken it into account. We have revised the statement accordingly. The new version provides a clearer and more comprehensive description of the treatment designs, specifying the activities encompassed by cultural treatment and providing a more precise outline of the comparisons considered. Thank you for your input in improving the clarity of our paper.
The new version of this paragraph would be the following: “Various treatment designs were included, all involving cultural treatment. Cultural treatment encompasses activities such as visiting museums, engaging in visual arts, listening to or performing music, dancing or watching dance performances, participating in performing arts, creating ceramics, attending readings or writing, or a combination of these activities. The treatment designs considered were cultural treatment compared to control, cultural treatment compared to another treatment type, and comparisons within different cultural treatment groups.”
Line 105: one of the authors (Author) — did you forget to replace (Author) by actual name, or does it mean otherwise? Same question for lines 127, 130, etc.
Thank you for pointing out the error. It was a mistake, and we have already fixed it by replacing "(Author)" with the correct initials of the authors.
Line 136: older adults — is there any threshold for this, like 65 or 70? If so, please mention this too.
Thank you for your question. In our study, when referring to "older adults," we consider individuals aged 60 and above. This age threshold is commonly used in research studies and is consistent with the definition of older adults in many contexts. We have updated the manuscript to include this clarification, along with a couple of references supporting this age range.
Line 177: significantly higher — why did you say ‘significantly higher’, instead of ‘higher’, or ‘high with 63.15%’?
Thank you for your feedback. We have revised the paragraph to provide a clearer and more concise description, eliminating the percentages among other changes.
Results
Tables 4: All Q values are close to zero, and their p-values are really high, which indicates all Q’s are no different from zero for each domain. Did you also compare one domain to another domain?
Thank you for your feedback. We appreciate your suggestion, and we have carefully considered it. Nevertheless, in order to streamline the manuscript and avoid redundancy, we have decided to remove tables 2, 3 and 4 from the paper. The information presented in these tables is already described in the text, ensuring that the key findings are effectively communicated. We value your input in enhancing the clarity of our research.
Discussion & Conclusions
As stated for Abstract, it would be better if you add some more statements about what you mean by ‘partial’ meta-analysis.
Thank you for your suggestion. We have taken your feedback into account and made the necessary revisions to address it. In both discussion and conclusion sections of the paper, we have included additional statements to clarify what we mean by 'partial' meta-analyses. These additions aim to provide a better understanding of the concept and its application within our study.
We have added some clarifications as follows:
Discussion: “Multiple partial MA’ were conducted, each focusing on studies related to a specific disease, a particular type of treatment, or a specific type of evaluation. This approach allowed for a more nuanced and detailed analysis of the benefits and outcomes associated with cultural interventions within each specific domain. By conducting these partial meta-analyses, the aim was to gain a deeper understanding of the effectiveness and impact of cultural interventions in targeted contexts, providing valuable insights into their outcomes in each of these specific areas. Once partial MA’ were conducted, some results supported the effectiveness of cultural interventions as treatment for people living with CI, but some further clarifications are necessary.
Conclusion: In conclusion, our results suggest that cultural interventions, due to their heterogeneity, cannot be assessed as a whole, but can highlight the wisdom of performing partial MA’ each focusing on studies pertaining to a specific disease, a particular type of treatment, or a specific type of evaluation to obtain sharper results, although these results must be interpreted with caution.”
We appreciate your valuable input in improving the clarity of our research.
Minor
I am not familiar with the format of reference numbering. Isn’t it better to use [1] instead of 1, which could be easily confused with actual value of 1? If it is required by the journal, I will accept it. Otherwise, please change the format of reference numbering.
Thank you for your suggestion. We appreciate your attention to detail. We will make the necessary changes to the format of reference numbering in accordance with your recommendation. We understand that using [1] instead of just 1 can help avoid confusion with actual values. We strive to ensure the clarity and accuracy of our references, and your feedback is valuable in achieving that. Thank you again for bringing this to our attention.
Reviewer 3 Report
The manuscript ˝Benefits of Cultural Activities on People with Cognitive Impairment: a Meta-Analysis˝ by Delfa-Lobato, et al. is a meta-analysis regarding the benefits of cultural activities on cognitive impairment connected to various diseases.
The review is well organized. Inclusion and exclusion criteria, as well as the search strategy, are well defined. The results are adequately statistically analysed and presented. The results showed that cultural interventions primarily contribute to a better quality of life and don't show significant improvement in cognitive impairment.
There are several comments and suggestions for the authors:
1. The reference numbers in the text were not placed in parentheses, which made it difficult to follow the text when reading.
2. In lines 127 and 130, it says "author" in brackets. I assume that there should be the initials of the co-authors who participated in the data extraction.
3. In lines 133 – 134, "and publication year" is written twice.
4. In the subtitle “3.2 Study characteristics”, the authors state that they included 26 studies in the analysis. They further state that 25 papers are in English and 1 in Spanish. For the paper in Spanish, they cite a new reference (ref. 56 - line 208) that was not mentioned when the distribution of the 26 papers by country was shown. That would mean they had 27 studies.
5. There is a problem with reference number 21, which is stated to be a systematic review and is mentioned several times during the discussion (lines 63, 500, 509, 525, 536, 562). In the reference list, Anonymous and 2021 are listed under number 21. This kind of literature citation is not acceptable as a reference. It should be supplemented, and the source from which it was taken should be stated.
6. In general, authors should check the reference list. The internet source should be stated in reference 2. There are parentheses at the beginning of several references (6, 12, 13).
Author Response
Thank you for your comments. Please find our responses below.
- The reference numbers in the text were not placed in parentheses, which made it difficult to follow the text when reading.
Thank you for your suggestion. We appreciate your attention to detail. We will make the necessary changes to the format of reference numbering in accordance with your recommendation. We understand that using [1] instead of just 1 can help avoid confusion with actual values. We strive to ensure the clarity and accuracy of our references, and your feedback is valuable in achieving that. Thank you again for bringing this to our attention.
- In lines 127 and 130, it says "author" in brackets. I assume that there should be the initials of the co-authors who participated in the data extraction.
Thank you for your feedback. We have already included the authors' initials as requested.
- In lines 133 – 134, "and publication year" is written twice.
Thank you for your feedback. We have removed the duplicated expression. Your input is appreciated and helps us improve the clarity of our paper.
- In the subtitle “3.2 Study characteristics”, the authors state that they included 26 studies in the analysis. They further state that 25 papers are in English and 1 in Spanish. For the paper in Spanish, they cite a new reference (ref. 56 - line 208) that was not mentioned when the distribution of the 26 papers by country was shown. That would mean they had 27 studies.
Thank you for bringing this to our attention. It was an oversight on our part. We apologize for the confusion caused. The correct number of included studies is indeed 26, and they are distributed among various countries as mentioned earlier. The reference citation (ref. 56 - line 208) for the Spanish paper was mistakenly included in the text, and it does not represent an additional study. We have revised the paragraph to ensure clarity and readability. We appreciate your feedback in helping us improve the accuracy of our paper.
- There is a problem with reference number 21, which is stated to be a systematic review and is mentioned several times during the discussion (lines 63, 500, 509, 525, 536, 562). In the reference list, Anonymous and 2021 are listed under number 21. This kind of literature citation is not acceptable as a reference. It should be supplemented, and the source from which it was taken should be stated.
Thank you for your guidance regarding reference 21. We have now completed it as suggested. Your input has been greatly appreciated.
- In general, authors should check the reference list. The internet source should be stated in reference 2. There are parentheses at the beginning of several references (6, 12, 13).
Thank you for pointing out the issues with the reference list. We have reviewed and corrected the internet source in reference 2 and removed the parentheses from references 6, 12, and 13. Your feedback is appreciated, and we strive for accuracy in our reference list.
Round 2
Reviewer 1 Report
Thank you for your effort and understanding of my comments and suggestions. The article now meets the journal's standards and can be published.